# NMR-Relaxometric Investigation of Mn(II)-Doped Polyoxometalates in Aqueous Solutions

**DOI:** 10.3390/ijms24087308

**Published:** 2023-04-15

**Authors:** Vladimir S. Korenev, Evgenia A. Burilova, Victoria V. Volchek, Enrico Benassi, Rustem R. Amirov, Maxim N. Sokolov, Pavel A. Abramov

**Affiliations:** 1Nikolaev Institute of Inorganic Chemistry SB RAS, 3 Akad. Lavrentiev Ave., Novosibirsk 630090, Russia; 2A.M. Butlerov Chemical Institute, Kazan Federal University, Kremlevskaya Str. 18, Kazan 420008, Russia; 3A.E. Arbuzov Institute of Organic and Physical Chemistry, FRC Kazan Scientific Center of RAS, Arbuzov Street 8, Kazan 420088, Russia; 4Faculty of Natural Sciences, Novosibirsk State University, 1 Pirogova Str., Novosibirsk 630090, Russia; 5Research School of Chemistry and Applied Biomedical Sciences, Tomsk Polytechnic University, Tomsk 634034, Russia

**Keywords:** Mn(II)-doped polyoxometalates, NMR-relaxometric studies, HPLC-ICP-AES

## Abstract

Solution behavior of K;_5_[(Mn(H_2_O))PW_11_O_39_]·7H_2_O (**1**), Na_3.66_(NH_4_)_4.74_H_3.1_[(Mn^II^(H_2_O))_2.75_(WO(H_2_O))_0.25_(α-B-SbW_9_O_33_)_2_]·27H_2_O (**2**), and Na_4.6_H_3.4_[(Mn^II^(H_2_O)_3_)_2_(WO_2_)_2_(β-B-TeW_9_O_33_)_2_]·19H_2_O (**3**) was studied with NMR-relaxometry and HPLC-ICP-AES (High Performance Liquid Chromatography coupled with Inductively Coupled Plasma Atomic Emission Spectroscopy). According to the data, the [(Mn(H_2_O))PW_11_O_39_]^5−^ Keggin-type anion is the most stable in water among the tested complexes, even in the presence of ethylenediaminetetraacetic acid (EDTA) or diethylenetriaminepentaacetic acid (DTPA). Aqueous solutions of **2** and **3** anions are less stable and contain other species resulting from dissociation of Mn^2+^. Quantum chemical calculations show the change in Mn^2+^ electronic state between [Mn(H_2_O)_6_]^2+^ and [(Mn(H_2_O))PW_11_O_39_]^5−^.

## 1. Introduction

One of the dominant research directions in the chemistry of nanomaterials is the development of new nanoprobes for medical imaging [1]. In this respect, polyoxometalates (POM) constitute promising candidates for development of such nanoprobes due to the following advantages: (i) biological activity (e.g., anti-RNA [2], anti-influenza [3], anti-COVID-19 [4]), which makes possible teranostic applications [5]; (ii) tendency for self-organization into hierarchically ordered nanostructures [6,7,8]; and (iii) suitable magnetic-relaxation characteristics of POM-based complexes [9,10]. From a structural point of view, spherical or nearly spherical nano-sized polyoxometalates have an internal cavity and pores on the surface, allowing an exchange between the mass of the solution and the internal cavity of the clusters [11]. The inherent high negative charge on the polyoxoanions allows control of their movement using electrophoresis [12,13].

In order to replace nephrotoxic Gd^3+^ in common magnetic resonance imaging (MRI) contrast agents currently in use, complexes and nanoparticles of other paramagnetic metals, in particular those of Mn^2+^, are currently under consideration as a viable alternative [14,15,16,17]. Though manganese complexes with lacunary polyoxometalates as ligands might be such candidates, there are few data on relevant characteristics of Mn(II)–POM complexes. In 2007, T_1_-relaxivity values were reported for “K_6_[MnSiW_11_O_39_]” (12.1 mM^−1^ s^−1^) and “K_8_[MnP_2_W_17_O_61_]” (4.7 mM^−1^ s^−1^), which turned out to be comparable with that of GdDTPA relaxivity [18]. The subsequent works on manganese-based contrast agents reported a significant improvement in contrast properties when manganese oxide colloids were included in nanoparticles [19]. The rotation correlation time contribution to the relaxivity can also be optimized by attachment, either covalent or non-covalent, of suitable paramagnetic complexes to various macromolecules such as human serum albumin (HAS), DNA, liposomes, or micelles. Another possibility is to increase the hydration number of the metal ion. In this regard, non-toxic, stable, and water-soluble materials based on organic matrices with inclusion complexes of manganese with polyoxometalate ligands can be envisaged as good candidates. As a prerequisite for such studies, the stability of possible candidates must be evaluated, first in water and subsequently in the presence of key components of biological liquids (salt buffers, HAS, etc.).

In this work, we chose three polyoxometalate complexes as such manganese-containing compounds: K;_5_[(Mn(H_2_O))PW_11_O_39_]·7H_2_O (**1**), Na_3.66_(NH_4_)_4.74_H_3.1_[(Mn^II^(H_2_O))_2.75_(WO(H_2_O))_0.25_(α-B-SbW_9_O_33_)_2_]·27H_2_O (**2**), and Na_4.6_H_3.4_[(Mn^II^(H_2_O)_3_)_2_(WO_2_)_2_(β-B-TeW_9_O_33_)_2_]·19H_2_O (**3**). Mn^2+^ in complex **1** is hydrated with 1H_2_O, in **2**—with 2 H_2_O, and in **3**—with 3H_2_O. Their solution behavior and magnetic-relaxation characteristics had not been reported yet. Therefore, the NMR relaxation study of their aqueous solutions, either alone or in the presence of additives, seems to be a necessary task.

The method of nuclear magnetic relaxation opens a possibility to detect changes in the composition of the coordination environment of paramagnetic metal ions [20]. We have shown [21] that the ratio of spin–spin and spin–lattice relaxation efficiency (relaxivity), *R*_2_/*R*_1_, in Mn(II) solutions correlates with the number of water molecules (*q*) in the first sphere of metal ions (Equation (1)):(1)$$\frac{R_{2}}{R_{1}}=\frac{{\left(1/T_{2}\right)}_{p}}{{\left(1/T_{1}\right)}_{p}}=\frac{{\left(1/T_{2M}\right)}_{dd}+{\left(1/T_{2M}\right)}_{con}}{{\left(1/T_{1M}\right)}_{dd}+{\left(1/T_{1M}\right)}_{con}}\approx \frac{{\left(1/T_{2M}\right)}_{dd}}{{\left(1/T_{1M}\right)}_{dd}}+\frac{{\left(1/T_{2M}\right)}_{con}}{{\left(1/T_{1M}\right)}_{dd}}=1.2+0.6q$$

In this way, it is easy to compare the values of *R*_2_/*R*_1_ ratio for a solution of a given Mn compound with the number of water molecules (six) coordinating with the [Mn(H_2_O)_6_]^2+^ cation and rapidly exchanging with other water molecules from the bulk solution (Appendix A). This ratio is 4.8 ± 0.2 for [Mn(H_2_O)_6_]^2+^ (*q* = 6), and it decreases to 1.2 ± 0.2 for the complex with diethylenetriaminepentaacetic acid, where all water molecules of the first coordination sphere are substituted with the donor atoms of the chelating ligand (*q* = 0). Thus, the relaxation rate ratio *R*_2_/*R*_1_ is an important tool for assessing the degree of substitution of water molecules in the first coordination sphere of manganese(II) ions with donor atoms both during complex formation and association processes [21,22,23,24].

In the Mn–POM complexes under consideration, manganese ions are embedded in a rigid polyoxometalate framework, and concentration of free [Mn(H_2_O)_6_]^2+^ in equilibrium may be quite low. Successful application of the NMR relaxation method to such objects implies occurrence of a rapid exchange of water molecules bound to Mn^2+^ with other H_2_O molecules in the bulk of the solution. In this case, it is necessary to make sure that the specified condition is satisfied for all paramagnetic ions in the cluster. This can be checked by analyzing the resistance of the Mn–POM complex to acid decomposition, which will convert all manganese ions into [Mn(H_2_O)_6_]^2+^ aqua ions, or by using a strong competing complexing agent (preferably, a chelator) to convert manganese ions into a complexonate. After that, according to the measured characteristics of solutions of aqua ions or the complexonate (which are well known for manganese(II) [25,26]), it is possible to estimate how stable the cluster is in a particular environment and whether the complete or partial release of Mn^2+^ is possible. For correct understanding of the information obtained in the NMR-relaxation method, it is also necessary to find out whether all Mn^2+^ ions in a complex actively contribute to the relaxation of protons of water molecules, that is, how many ions should be taken into account when calculating the “active” concentration of paramagnetic ions included in the equation for calculating the parameters R_1_ and R_2_.

In this work, we studied the stability of compounds **1**–**3** in water with NMR-relaxation and HPLC-ICP-AES techniques.

## 2. Results and Discussion

### 2.1. Synthesis and Characterization of Mn-Containing Complexes

As was previously reported, addition of Mn^2+^, as with any other 3d M^2+^ or M^3+^ cations to [PW_11_O_39_]^7−^, either prepared ad hoc or generated in situ, leads to incorporation of Mn^2+^ inside the lacune of the monolacunary Keggin-type (the best known structural form for heteropoly acids of [XM_12_O_40_]^n−^ general formula, where X is the heteroatom (e.g., P^V^, Si^IV^, etc), M is the addendum atom (e.g., Mo^VI^ or W^VI^)) anion backbone producing [Mn(H_2_O)PW_11_O_39_]^5−^ in high yield [27]. In our case, we isolated the complex as K_5_[Mn(H_2_O)PW_11_O_39_]·7H_2_O (**1**). The purity of the product was confirmed using total elemental analysis and chromatography. In the crystal structure, {Mn(H_2_O)}^2+^ unit is disordered over all 12 metal positions, which makes single crystal X-ray diffraction (SCXRD) analysis uninformative [27]. The idealized structure of [Mn(H_2_O)PW_11_O_39_]^5−^ is shown in Figure 1a.

Preparations of **2** and **3** were based on the protocols reported by Krebs et al. [28]. We have found, however, that reaction between Na_9_[SbW_9_O_33_]·19.5H_2_O and MnCl_2_·4H_2_O in water produces a mixture of [(Mn^II^(H_2_O))_3−x_(WO(H_2_O))_x_(α-B-SbW_9_O_33_)_2_]^n−^ anions (Figure 1b) instead of the pure [(Mn^II^(H_2_O))_3_(α-B-SbW_9_O_33_)_2_]^12−^ reported by the authors. The solid product was isolated as crystalline material upon air evaporation. According to elemental analysis, its composition was formulated as Na_3.66_(NH_4_)_4.74_H_3.1_[(Mn^II^(H_2_O))_2.75_(WO(H_2_O))_0.25_(α-B-SbW_9_O_33_)_2_]·27H_2_O (**2**). Formation of such structures where {W(O)}^4+^ and {M(H_2_O)}^n+^ are competitively sandwiched between to {EW_9_} moieties is not uncommon [29]. According to SCXRD, **2** crystallizes in monoclinic crystal system (*P*2_1_/m space group) with the following unit cell parameters: a = 12.9133, b = 19.0752, and c = 16.7256, β 101.216°. The structure of the anion is shown in Figure 1b. The structure of [(Mn^II^(H_2_O))_3_(α-B-SbW_9_O_33_)_2_]^12−^ has been reported [30,31,32,33]; that is why we obtained only a satisfactory structural model of **2**, which is enough to extract theoretical data for XRPD analysis. This was used to confirm the phase purity according to XRPD (Appendix A). 

The synthesis of **3** was carried out according to the procedure reported for Na_8_[(Mn^II^(H_2_O)_3_)_2_(Mn^II^(H_2_O)_2_)_2_(TeW_9_O_33_)_2_]·34H_2_O [28]. The yellow crystalline product was isolated after slow (several days) crystallization from the reaction solution in air. Poorly diffracted single crystals of **3** were studied by SCXRD. The complex crystalizes in triclinic crystal system (*P*-1 space group) with the following unit cell: a = 13.3405 Å, b = 14.1672 Å, c = 15.0359 Å, α = 68.394°, β = 71.165°, and γ = 73.784°. The refinement gives Krebs-type [(Mn^II^(H_2_O)_3_)_2_(WO_2_)_2_(B-β-TeW_9_O_33_)_2_]^8−^ anion (Figure 1c). This structure is the same as [(Mn^II^(H_2_O)_3_)_2_(WO_2_)_2_(BiW_9_O_33_)_2_]^10−^ [28]; that is why we obtained a satisfactory structural model only of **3**. The product purity was confirmed usinf full elemental analysis and chromatography. The crystalline material is unstable upon air drying and quickly becomes amorphous.

### 2.2. Quantum-Chemical Calculations of Mn^2+^ Electronic States 

Complexes **1**–**3** are all bright yellow, despite the fact that [Mn(H_2_O)_6_]^2+^ is almost colorless (very pale pink) because d–d transitions for high-spin d^5^ ions in O_h_ field are both spin- and Laporte-forbidden. When Mn^2+^ enters the lacunary site, the local symmetry becomes D_4h_ (idealized) or even C_s_ (exact). To confirm this distortion, we calculated the idealized structure of [(Mn(H_2_O))PW_11_O_39_]^5−^, which is shown in Figure 1a. Upon geometry optimization, carried out at DFT B3LYP[GD3BJ]/6-311++G** (for H, P, O, and Mn) and LANL2TZ(f) +ECP (for W) levels (see Computational Details), one of the two H atoms of the water molecule interacted with an O atom of the cluster (H-O_clus_ = 2.268 Å), whereas the coordination of the Mn^2+^ was found to be distorted prolate octahedron. The equatorial position is occupied by four O atoms belonging to the W–O cluster (Mn-O_clus_ = 1.876, 1.914, 1.945, 2.073 Å), whereas the apical positions are occupied by an O atom of the phosphate ion (Mn-O_phos_ = 2.166 Å) and by an O atom of the water molecule (Mn-O_water_ = 2.259 Å); O_phos_, Mn, and O_water_ are not aligned, describing a bond angle of 169^o^. This C_s_ distortion permits allowed d–d transitions for Mn^2+^ in the UV–Vis spectrum. In addition, charge transfer from Mn(II) to W(VI) bands are also allowed. Time-dependent density-functional theory (TD-DFT) calculations were performed using ω-B97X[D] functional and the same basis sets used for the geometry optimisation to study the nature of the yellow coloration associated with Mn^2+^ into the cavity of [PW_11_O_39_]^7−^. Five allowed electronic transitions were identified between 2 and 4 eV; in particular, an intense peak at 2.198 eV, two low-intensity peaks at 2.530 and 2.579 eV, and two middle-intensity peaks at 2.601 and 3.625 eV. A good match between predicted and experimental bands was found, both for transition energies and intensities. The UV–Vis absorption spectrum of **1** in water at neutral pH is poorly informative, as it does not show any clear structure; in particular, the absorbance is negligible above 500 nm whereas a rapid, monotonic increase was observed below 500 nm probably due to charge transfer processes (Figure 2 (left)). On the contrary, the diffuse reflectance (DR) spectrum in solid phase shows two main bands at 2.21 eV and 3.66 eV; the former has a shoulder about 2.55 eV (Figure 2 (right)).

In order to clarify the complex nature of the three more intense electronic transitions, the Natural Transition Orbital (NTO) analysis was applied, using the same functional and basis sets used for the TD-DFT calculations (Appendix A). The two transitions at 2.198 eV and 2.601 eV show a strong charge transfer character; Mn’s *d* orbitals are involved as acceptors of electron density. On the other hand, the NTO analysis of the electronic transition at 3.625 eV indicates the involvement of Mn’s *d* orbitals both as donors and acceptors. We can therefore conclude that the color of this Mn^II^-containing compound is associated to electronic transitions, whose oscillator strength is non null thanks to the distorted coordination of the metal; despite the common involvement of Mn’s *d* orbitals, these transitions are different in nature, *viz.* they have a charge transfer character (at 2.198 eV), a delocalization character (at 2.601 eV), and a local excitation character (at 3.625 eV).

### 2.3. HPLC-ICP-AES

HPCL-ICP-AES hyphenated technique [34] is a powerful tool to study the reactivity/stability/evolution of different polyoxometalates in aqueous and organic solutions [35,36,37]. In the present research, this technique was a good complement for NMR-relaxation studies. The HPLC-ICP-AES chromatogram of K_5_[(Mn(H_2_O))PW_11_O_39_]·7H_2_O shows a single peak (t_R_ = 4.6 min) with atomic ratio W:Mn = 11.1 ± 0.5. The data obtained from HPLC-ICP-AES (Figure 3) were used to calculate the atomic ratios of the elements (Appendix A). The phosphorus content was close to the detection limit of the element by HPLC-ICP-AES; therefore, the intensities of signals from P are significantly lower than Mn and W and cannot be adequately treated. In order to confirm the peak purity of the observed signal, we calculated atomic ratio at each point of the HPLC-ICP-AES chromatogram for all pairs of detected spectral lines (Figure 3). Any impurity eluted along with the main peak will cause a deviation from the horizontal line of the calculated atomic ratio. The atomic ratios of the pairs of spectral lines W/Mn were constant during the entire HPLC-ICP-AES analysis. Therefore, it can be argued that the peak does not contain impurities, and there is only one individual species in the aqueous solution.

The HPLC-ICP-AES chromatogram of a solution of Na_3.66_(NH_4_)_4.74_H_3.1_[(Mn^II^(H_2_O))_2.75_(WO(H_2_O))_0.25_(α-B-SbW_9_O_33_)_2_]·27H_2_O in water is shown in Figure 4. The first peak (1) with W:Mn = 6.5 ± 0.5, W:Sb = 9.1 ± 0.6 and the second (2) Mn-free peak corresponding to the *antimonotungstate* (W:Sb = 9.3 ± 0.5) overlap, which may indicate an excess of tungsten or possible sample decomposition under the separation conditions. These data show that [(Mn^II^(H_2_O))_3−x_(WO(H_2_O))_x_(α-B-SbW_9_O_33_)_2_]^11.5−^ is not a stable species in water and equilibrate with Mn-free species (peak №2) and W- and Sb-free species (Appendix A) that are not retained on the column and elute with a dead volume. Possible explanation for this effect could be the presence of the positively charged species (e.g., [Mn(H_2_O)_6_]^2+^) with no affinity for the HPLC-column in the absence of an ion-pair reagent for cationic species. The same peak is also observed in the chromatograms of the complexes **1** and **3**. Unfortunately, it is not possible to calculate the atomic ratio for the detected peak as it contains only Mn spectral lines.

The HPLC-ICP-AES chromatogram of a solution of Na_4.6_H_3.4_[(Mn^II^(H_2_O)_3_)_2_(WO_2_)_2_(β-B-TeW_9_O_33_)_2_]·19H_2_O in water at “natural” pH shows two baseline-unresolved peaks (Figure 5). The first peak (1) was attributed to [(Mn^II^(H_2_O)_3_)_2_(WO_2_)_2_(β-B-TeW_9_O_33_)_2_]^8−^ due to the following atomic ratios of the elements: W:Mn = 10.1 ± 0.4, W:Te = 9.9 ± 0.5, and Mn:Te = 1.1 ± 0.2. The second peak (2) can be attributed to a manganese-free tellurotungstate with atomic ratio W:Te = 9.2 ± 0.5. This means that the complex dissociates with the formation of free “{TeW_9_O_33_}”, which can be a monomeric or “dimeric” (such as [Te_2_W_18_O_62_(OH)_2_]^10−^ [38]) species.

Thus, the complexes can be ranked, according to their stability in water, in the following order: **1** > **3** >> **2**. showing behavior of **1**–**3** in water according to NMR relaxation data.

### 2.4. Behavior of ***1**–**3*** in Water According to NMR Relaxation Data

As the first step, we studied influence of the medium acidity on the magnetic relaxation characteristics of **1**–**3** in an aqueous solution. In order to correctly calculate the relaxivity parameters, R_1_ and R_2_ relaxation times of a solution of **3** (0.1 mM) were measured. The relaxivities and the R_2_/R_1_ ratio were calculated with Equation (3) taking into account two possible Mn species in solution, with four ([(Mn^II^(H_2_O)_3_)_2_(Mn(H_2_O)_2_)_2_(β-B-TeW_9_O_33_)_2_]^8−^) or with two ([(Mn^II^(H_2_O)_3_)_2_(WO_2_)_2_(β-B-TeW_9_O_33_)_2_]^8−^) manganese ions, that is, C_Mn_ = 0.4 or 0.2 mM, respectively (Figure 6).

It follows from the obtained data that in a strongly acidic medium, the R_2_/R_1_ ratio approaches the values of 4.8–5.0, which are characteristic of [Mn(H_2_O)_6_]^2+^ (compare curves 1–3, Figure 6b). Consequently, compound **3** is unstable to acid and already at pH < 4 undergoes decomposition releasing Mn as [Mn(H_2_O)_6_]^2+^. At the same time, the relaxivity values R_1_ (7900–8000 M^−1^s^−1^), characteristic of [Mn(H_2_O)_6_]^2+^ in Figure 6a, are observed for curve 2 obtained under the assumption that there are only two magnetically active manganese ions in compound **3**. (Measurement data for spin–spin relaxivity are shown in Appendix A. For manganese aqua ions, the R_2_ values are 38,000–39,000 M^−1^s^−1^.) In this case, the complete coincidence of curves 1 and 2 in Figure 6b is not surprising, since the R_2_/R_1_ ratio does not depend on the chosen concentration of manganese ions. Thus, according to the measured proton NMR-relaxation rates, it can be concluded that in compound **3**, there are only two manganese ions catalyzing the relaxation of protons of water molecules. Compound **3** can be stable in water in the pH range of 4–8; in more acidic solutions, it decomposes with the release of two manganese(II) aqua ions. This confirms the XRD data on the presence of only two manganese ions in compound **3**.

When analyzing the relaxivities R_1_ and R_2_ found for solutions of **3**, and their ratios R_2_/R_1_, two interesting points arose. The first is that in the pH range 4–8 R_1_ is almost two-fold higher than R_1_ [Mn(H_2_O)_6_]^2+^ (curves 2 and 3, Figure 6a). This can be explained by the fact that the proton spin–lattice relaxation in Mn(II) solutions is controlled by the correlation rotation time τ_R_ [22,39]. It is well known that the relaxation rate can significantly increase when manganese ions bind to large objects such as polymers, macrocycles, proteins, or nanoparticles, as already observed in previous works [21,22].

The second point is that the value of the ratio R_2_/R_1_~1.5–1.6 at its own pH (5.5) corresponds to the presence of no more than one water molecule in the internal coordination sphere of each manganese ion. This contradicts the formula composition, according to which compound **3** contains manganese ions associated with three water molecules. Possibly, this effect is a consequence of the structural features of cluster **3**, and only two water molecules associated with two magnetically active manganese ions are active in relation to exchange with other water molecules in the bulk of the solution. The structure in solution can, of course, be different from that in crystal, even if the molar ration of building blocks is retained. These explanations, of course, are based solely on NMR-relaxation data and need to be verified with other methods.

With respect to **1** and **2**, similar experiments were carried out to check the dependence of their relaxivities R_1,2_ and the ratio R_2_/R_1_ on pH, which are shown in Figure 7 and Appendix A in comparison with the data for **3**.

From Figure 7, it can be seen that for **1** the coordination sphere of Mn^2+^ is the most stable: the R_1_ values and the R_2_/R_1_ ratio are constant over a wide pH range. The R_2_/R_1_ value of 1.6±0.1 is close to the value of 1.8, corresponding to the presence of one water molecule bound to Mn^2+^, which fully corresponds to the chemical composition of the polyoxoanion. At the same time, over the pH range of 2.5–8, the relaxivity values are not only significantly lower than the R_1_ values for **3**, but they are also lower than the values for [Mn(H_2_O)_6_]^2+^. This demonstrates again that relaxivity is a function of many factors, including the number of coordinated water molecules and the correlation time [20]. Despite the fact that compound **1** remains stable in water up to pH 2.5, further acidification finally leads to a change in the degree of hydration of manganese ions, as indicated by an increase in the R_2_/R_1_ ratio, accompanied by the approach of the R_1_ and R_2_ values to the relaxivity values of [Mn(H_2_O)_6_]^2+^ (Figure 7 and Appendix A). Thus, we can also propose the change in the original structure of **1**. Possibly, at pH < 2.5 the anion of **1** decomposes to yield plenary Keggin phosphotungstate [PW_12_O_40_]^3−^ and [Mn(H_2_O)_6_]^2+^.

For **2**, which in ideal case contains three identical manganese atoms, the values of the ratio R_2_/R_1_ = 1.75–1.95 are consistent with the structural data that each of the manganese cations is bound to one water molecule. Meanwhile, the resistance of a solution of compound **2** to acidification is significantly inferior to that observed for **3** and especially for **1**, since the R_2_/R_1_ ratio begins to increase already at pH below 5 (Figure 7b). It is interesting to note that this process has almost no effect on the R_1_ values, which are initially quite close to R_1_ for [Mn(H_2_O)_6_]^2+^, but the spin–spin relaxivity values increase, approaching the R_2_ values for [Mn(H_2_O)_6_]^2+^ (Appendix A). This is in accordance with HPLC-ICP-AES data (Appendix A).

Conclusions from the analysis of the data obtained in the NMR relaxation study of solutions of compounds **1** and **2** in water are generally consistent with the structure of these compounds obtained by X-ray diffraction analysis [28]. At the same time, according to these data, it was possible to estimate the number of manganese atoms in compound **3** and to collect additional information concerning hydrolytic stability of polyoxocomplexes.

### 2.5. Effect of the Cationic Polymer on Compound ***3*** in Water

It was previously shown that the interaction of the cationic form of polyethyleneimine (PEI) with anionic complexes of manganese with EDTA or DTPA in an acidic medium leads to an increase in relaxivity due to the formation of complex-polymer associates [25,26]. The selected polyoxoanions are anionic in nature; their behavior in solution was therefore checked in the presence of a cationic polyelectrolyte. For this purpose, the effect of polyethyleneimine (PEI) additives on the magnetic relaxation parameters of a solution of compound **3** was studied. When PEI is introduced into a solution of **3**, a significant change in the behavior of the pH dependences of both the R_1_ and R_2_ values and the R_2_/R_1_ ratio is observed (Figure 8 and Appendix A). The region of elevated R_1_ relaxivity values, which was explained above by the slow rotation of the cluster, disappeared, and, based on the high R_2_/R_1_ ratio values (4.75–4.95), almost up to pH 7 manganese ions are present as aqua ions.

In general, the profile of curves 2 in Figure 8 and Appendix A is similar to that of curves 3 depicted in our previous study of the state of manganese(II) in polymer solutions [22]. The decrease in relaxivity above pH 7 and the decrease in the hydration of manganese ions are explained by the binding of these ions to numerous amino groups in the branched polymer, which loses its cationic nature upon deprotonation of its ammonium groups. Thus, it is obvious that this polyoxoanion will be extremely unstable in solutions containing cationic polyelectrolytes.

### 2.6. Stability of Compounds ***1**–**3*** in Complexone Solutions

When screening compounds for use in biosystems, considerable attention should be paid to their stability to prevent degradation and poisoning of organisms by possible toxic products, including multiply charged metal ions. For this purpose, for example, a complex of gadolinium (or manganese) with polydentate chelators such as diethylenetriaminepentaacetic acid (DTPA) is used as an MRI contrast agent. Thus, by studying the behavior of a polyoxoanion in a complexone solution, one can evaluate its relative stability and predict its prospects for biomedical applications.

Previously, we studied the state of stable complexes of Mn^2+^ with EDTA and DTPA in solutions of the cationic PEI polymer, where pH regions of the formation of Mn-L(EDTA or DTPA)-PEI ternary compounds were found. It was interesting to study the possibility of {Mn(H_2_O)_n_}_polyoxoanion_-L(EDTA or DTPA) ternary systems. To do this, the experiments were organized in such a way that a variable amount of a complexone (less and more than one equivalent) was added to the solutions of **1**–**3**, while maintaining the acidity of the solutions constant. For comparison, a similar experiment was also carried out with a manganese salt solution of the same concentration.

At the first stage, EDTA was chosen as a complexing agent, and the results are shown in Figure 9, Figure 10, Appendix A. Using compound **3** as an example, we tested the polyoxoanion behavior in the presence of EDTA in a wide range of pH. Mn^2+^ forms strong complexes with EDTA in solutions (lgK_1_ = 13.6 in 0.1 M KNO_3_, NIST Standard Reference Database 46 (Critically Selected Stability Constants of Metal Complexes), Version 7.0), and at pH > 3 the relaxivity does not change, which indicates the completion of the complex formation process.

Of those presented in Figure 10 and Appendix A, the relaxivity curves R_1_ and the ratio R_2_/R_1_ for [Mn(H_2_O)_6_]^2+^ and for **3** almost completely coincide with one another in an excess of EDTA. Thus, in contrast to compound **2**, the complete transition of manganese ions from polyoxoanion **3** into the complexonate in solution with pH 5.5 needs more than equivalent quantity of EDTA.

Compound **2** expectably turned out to be unstable in a complexone solution, and when an equimolar amount of EDTA is added, a kink is observed in the relaxation curves with a constant relaxation value (curves 2 in Figure 11 and Appendix A) corresponding to the formation of manganese complexonate (curves 4 in Figure 10 and Appendix A). This pattern is typical for the formation of a strong complex with a metal:ligand composition of 1:1.

For compound **1**, however, the addition of EDTA to a solution with pH 5.5 did not reveal any changes, since it turned out that the values of both relaxivities and the ratio R_2_/R_1_ of [(Mn(H_2_O))PW_11_O_39_]^5−^ and manganese complexonate completely coincide (curves 1 and 4 in Figure 10 and Appendix A at an EDTA content above 0.4 mM). This result is unexpected and can be explained with the fact that in both compounds in the first sphere of the manganese ion there are five atoms of the coordinated environment (polyoxometalate in **1** or O and N atoms from EDTA), and the sixth place is occupied by a water molecule. Thus, to elucidate the possibility of the transfer of Mn^2+^ from **1** to the complexonate, it is necessary to use a ligand, the complex that will have different relaxivities than the complex with EDTA. This requirement was met with DTPA, and the results are presented in Figure 11 and Appendix A.

Addition of DTPA to solutions of Mn^2+^ indicates the formation of a 1:1 complex (Figure 11a and Appendix A), lg ML = 15.6 (NIST Standard Reference Database 46 (Critically Selected Stability Constants of Metal Complexes), Version 7.0). It can be seen that, for complexes of Mn^2+^ with DTPA, the relaxivities R_1_ and R_2_ are 1200 and 1600 M^−1^s^−1^, respectively, which is noticeably lower than for **1** (3000 and 4500 M^−1^s^−1^). From all these data, it can be seen that when DTPA is added to a solution of **1**, Mn^2+^ transfer from [(Mn(H_2_O))PW_11_O_39_]^5−^ to the complexonate is negligible. Therefore, the tight fixation of Mn^2+^ in the structure of [(Mn(H_2_O))PW_11_O_39_]^5−^ can be a starting point for the design of paramagnetic markers resistant to competitive ligands (for example, MRI contrast agents). In this regard, the only disadvantage of compound **1** should be considered as a relatively low relaxation efficiency, which is comparable to that obtained for the Mn(II) complex with EDTA; however, it exceeds the parameters for DTPA complex. For compound **3**, a decrease in relaxivities is also observed, and at the maximum content of DTPA, the parameters turn out to be close to those found for manganese complexonate (curves 3 in Figure 11a and Appendix A). Thus, complex **3** is destroyed by DTPA, and, despite its high spin–lattice relaxation (11,000 M^−1^s^−1^), this structure is not of interest as a platform for creating paramagnetic biomarkers.

## 3. Materials and Methods

### 3.1. General Information

Manganese(II) chloride (99%, Khimreaktiv, Moscow, Russia), EDTA (99%, Merck Millipore, Burlington, MA, USA), DTPA (99%, Merck Millipore, Burlington, MA, USA), branched polyethyleneimine (PEI) as 50% aqueous solution (M.n. 60000, Sigma-Aldrich Co, St Louis, MO, USA), sodium hydroxide (99%, Khimreaktiv, Moscow, Russia), and hydrochloride acid (99%, Khimreaktiv, Moscow, Russia) were used. Ultra-purified water (18.2 MΩcm resistivity at 25 °C) was produced from Direct-Q 5 UV equipment (Millipore S.A.S. 67120 Molsheim-France). Experiments and measurements were conducted at 298 K. The temperature was maintained using Haake DC10 (Thermo Fisher Scientific GmbH, Karlsruhe, Germany) cryo thermostat. PEI concentration is expressed in its “monomeric” units relative to the respective molecular weight. Water content was determined with thermal gravimetric analysis using TG Analyzer TG 209 F3 Tarsus (NETZSCH). Infrared spectra (4000–400 cm^−1^) were recorded on a Scimitar FTS 2000 spectrophotometer in KBr pressed pellets.

### 3.2. Synthesis

**Synthesis of K_5_[(Mn^II^(H_2_O))PW_11_O_39_]·7H_2_O** (**1**): K_7_[PW_11_O_39_]·14H_2_O (10 g, 3.1 mmol) was dissolved in water (5 mL) under gentle heating. Solid MnCl_2_·4H_2_O (0.615 g, 3.1 mmol) was added leading to a dark red solution. The reaction mixture was stirred for 5 h at room temperature, and solid KCl (2 g, 26.8 mmol) was added. Pale yellow precipitate formed during 1 h was isolated using filtration; washed with ethanol, diethyl ether; and air dried, yield 87%. IR (KBr, ν/cm^−1^, Appendix A) was: 1624 (m); 1080 (m); 1051 (s); 976 (s, sh); 957 (vs); 891 s; 831 s, sh; 804 vs, br; 763 s; 715 s, br; 592 m; 511 m; and 484 m. TGA: An average loss of ca. 8 water molecules between room temperature and 220 °C corresponding to crystallization and coordinated water molecules.

**Synthesis of Na_3.66_(NH_4_)_4.74_H_3.1_[(Mn^II^(H_2_O))_2.75_(WO(H_2_O))_0.25_(SbW_9_O_33_)_2_]·27H_2_O** (**2**): Compound **2** was prepared according to the published procedure [28]. A dark orange microcrystalline product was collected using filtration; washed with ethanol, diethyl ether; and air dried, yield 92%. IR (KBr, ν/cm^−1^, Appendix A) was: 1632 (m); 1416 (m); 930 (s); 862 (vs); 772 (s, sh); 714 (vs); 690 (vs); 637 (s, sh); 509 (s); 465 (s); and 436 (s). TGA: An average loss of ca. 30 water molecules between room temperature and 220 °C corresponding to crystallization and coordinated water molecules.

**Synthesis of Na_4.6_H_3.4_[(Mn^II^(H_2_O)_3_)_2_(WO_2_)_2_(B-β-TeW_9_O_33_)_2_]·19H_2_O** (**3**): *Solution 1:* TeO_2_ (0.22 g, 1.34 mmol) was dissolved under gentle heating in 10 M NaOH solution (0.4 g in 1 mL of water) and diluted with 10 mL of water. *Solution 2:* Na_2_WO_4_·2H_2_O (3.96 g, 12.01 mmol) was dissolved in a mixture of water (20 mL) and 12 M HNO_3_ (0.7 mL) and heated to 75 °C. Solution 1 was added dropwise to solution 2, and Na_2_CO_3_ (0.4 g, 4.8 mmol) was added after. A solution of MnCl_2_·4H_2_O (0.532 g, 2.68 mmol) in water (10 mL) was added slowly to the reaction mixture leading to a cloudy pale yellow solution with pH~9.3. The pH was set to 3 with dropwise addition of a concentrated HNO_3_ solution, resulting clear deep yellow mixture that was heated for 1 h at 80 °C, filtered, and allowed to cool to ambient temperature. A yellow crystalline product was obtained after several days; isolated; washed with ethanol, diethyl ether; and air dried, yield 35%. IR (KBr, ν/cm^−1^, Appendix A) was: 1626 m; 984 m, sh; 972 s; 880 s, sh; 839 vs; 779 vs; 741 s; 691 s; 650 s, br; 505 m; and 478 m. TGA: An average loss of ca. 25 water molecules between room temperature and 220 °C corresponding to crystallization and coordinated water molecules.

### 3.3. NMR-Relaxation

Proton relaxation times *T*_1,2_ were measured using pulsed NMR-relaxometer Minispec MQ20 (Bruker) with operational frequency 19.65 MHz applying standard radio frequency pulse sequences: inversion-recovery method for spin–lattice relaxation time (*T*_1_), and Carr–Purcell sequence modified by Meiboom–Gill for spin–spin relaxation time (*T*_2_) with measuring accuracy better than 3%. The temperature was maintained using Haake DC10 (Thermo Electron) cryo thermostat.

The experimentally measured relaxation times (*T*_2_)*_obs_*, s, were inverted into the relaxation rates (1/*T*_2_)*_obs_*, s^−1^. The relaxation rate is the sum of the two main contributions: the relaxation of protons in water (1/*T*_2_)*_d_* (diamagnetic component) and the relaxation of the protons around the paramagnetic ion (1/*T*_2_)*_p_* (paramagnetic component):(2)$${\left(\frac{1}{T_{1,2}}\right)}_{obs}={\left(\frac{1}{T_{1,2}}\right)}_{p}+{\left(\frac{1}{T_{1,2}}\right)}_{d}$$

The paramagnetic component, (1/*T*_1,2_)*_p_*, was calculated according to Equation (2) as the difference between the measured relaxation rate (1/*T*_1,2_)*_obs_* (measured for Mn-containing solutions) and the diamagnetic component (1/*T*_1,2_)*_d_* (for diluted aqueous solutions is  equal to 0.4 s^−1^).

Using manganese concentration, *C*, the paramagnetic component, (1/*T*_1,2_)*_p_*, was converted into relaxivity *R*_1,2_, *M*^−1^s^−1^, according to Equation (3):(3)$$R_{1,2}=\left(\frac{1}{C_{M}T_{1,2p}}\right)$$

### 3.4. Diffuse Reflectance Spectra 

DR spectra were measured on a setup that consists of a Kolibri-2 spectrometer (VMK Optoelektronica, Novosibirsk, Russia), fiber optic cable QR-400-7 (Ocean Optics, Orlando, FL, USA), and deuterium–tungsten lamp AvaLight-DHS (Avantes, Apeldoorn, The Netherlands) [40]. The reference of 100% reflectance was BaSO_4_ powder.

### 3.5. X-ray Diffraction on Single Crystals

Crystallographic data and refinement details are given in Appendix A. The diffraction data for **1** and **2** were collected on a Bruker D8 Venture diffractometer with a CMOS PHOTON III detector and IµS 3.0 source (Mo Kα radiation, λ = 0.71073 Å) at 150 K. The φ- and ω-scan techniques were employed. Absorption correction was applied by SADABS (Bruker Apex3 software suite: Apex3, SADABS-2016/2 and SAINT, version 2018.7-2; Bruker AXS Inc.: Madison, WI, USA, 2017). Structures were solved using SHELXT [41] and refined with full-matrix least-squares treatment against |F|^2^ in anisotropic approximation with SHELX 2014/7 [42] in ShelXle program [43].

### 3.6. X-ray Powder Diffraction

X-ray powder diffraction patterns were measured on a Bruker D8 Advance diffractometer using LynxEye XE T discriminated CuKα radiation. Samples were layered on a flat plastic specimen holder.

### 3.7. HPLC-ICP-AES

Separation was performed with HPLC system Milichrom A-02 (EcoNova, Novosibirsk, Russia) equipped with a two-beam spectrophotometric detector at the wavelength range of 190−360 nm in ion-pair mode of reversed phase chromatography (ProntoSIL 120-5-C18AQ, 2 × 75 mm), eluents: A—0.02% tetrabutylammonium hydroxide; B—acetonitrile. Gradient elution with gradual increase in acetonitrile concentration was employed to resolve the species. ICP-AES (inductively coupled plasma atomic emission spectrometry) spectrometer iCap 6500 Duo (Thermo Scientific, Waltham, MA, USA) with a concentric nebulizer was applied as detector in hyphenated HPLC-ICP-AES. For the element detection of Mn 259.3 nm, Mn 260.5 nm, Mn 279.4 nm, W 209.8 nm, W 229.4 nm, and W 239.7 nm, spectral lines were selected. In order to eliminate plasma quenching, we diluted the liquid coming out of the column into the spray chamber with deionized water. The steady state of the plasma and the optimal values of analytical signals were finally achieved at the eluent flow rate of 0.25 mL min^−1^ and the eluent velocity of 3 mL min^−1^ (peristaltic pump speed—75 rpm). Element contents in samples were determined using a high-resolution spectrometer iCAP-6500 Duo (Thermo Scientific). A typical sample amount of 10 mg was dissolved in deionized water (R ≈ 18 MΩ) and diluted to the volume of 10 mL prior to the ICP-AES experiment. The working parameters of the ICP-AES system include: power supply—1150 W, nebulizer argon flow rate—0.70 L min^−1^, auxiliary—0.50 L min^−1^, and cooling—12 L min^−1^. All measurements were performed in three replicates. The data acquisition and processing were carried out with iTEVA 2.0.0.39 (Thermo Scientific, USA) software. 

### 3.8. Quantum-Chemical Calculations

The molecular geometry of [Mn(H_2_O)PW_11_O_39_]^5−^ was fully optimised, with no symmetry constrain in the gas phase, used at the unrestricted Density Functional Theory (DFT) level. The initial guess of the atomic spatial coordinate was taken from the experimental structure. The hybrid functional B3LYP [44,45] coupled with the Pople triple-ζ basis set 6-311++G** for H, P, O, and Mn, and the LANL2TZ(f) triple-ζ basis was set on the W valence shell with the LANL2TZ(f) effective core potential on the core [46,47,48]. The D3 version of Grimme’s dispersion with Becke–Johnson damping [49] was included. 

The vibrational frequencies and thermochemical values were computed for the optimised geometries at the same levels of theory, within the harmonic approximation, at *T* = 298.15 K and *p* = 1 atm; no imaginary frequencies were found, indicating that the stationary points were “genuine” minima of the potential energy hypersurface. 

The UV–Vis absorption spectra for the equilibrium geometries were calculated at time dependent density functional theory (TD-DFT) level, accounting for D_1_ → D_1+*n*_ (*n* = 1 to 20). The nature of the vertical excited electronic state was analyzed. This investigation was performed by employing the long-range corrected functional ω-B97X[D] coupled with the same bases set used for optimization. The Natural Transition Orbital (NTO) analysis [50] was performed for the transitions of interest. 

The atomic charge population analysis, electric multiple moments, electronic density, and electrostatic potential were also computed within the Mulliken partition scheme. 

The integration grid was set to 250 radial shells and 974 angular points. The convergence criteria for the self-consistent field were set to 10^−12^ for the RMS change in the density matrix and 10^−10^ for the maximum change in the density matrix. The convergence criteria for optimizations were set to 2·10^−6^ a.u. for the maximum force, 10^−6^ a.u. for the RMS force, 6·10^−6^ a.u. for the maximum displacement, and 4·10^−6^ a.u. for the RMS displacement.

All calculations were performed using the GAUSSIAN G16.A01 package [51].

## 4. Conclusions

The NMR-relaxation method and HPLC-ICP-AES techniques were used to characterize aqueous solutions of manganese(II)-containing polyoxometalates **1**–**3**. To elucidate the hydration state of manganese ions in polyoxoanions, we used an approach based on the analysis of the R_2_/R_1_ ratio of relaxivities. According to the values of spin–lattice relaxation, compounds can be ranked (in descending order): **3** > **2** > **1**. The stability of aqueous solutions of compounds **1**–**3** was studied in a wide pH range; the widest range of hydrolytic stability (pH 2.5–8) was found for compound **1**. Polyoxometalate **3** is unstable in the presence of a cationic polyethyleneimine polymer. In the presence of complexones (EDTA and DTPA), compounds show a noticeable difference in stability (in decreasing order): **1** > **3** >> **2**. Quantum chemical calculations were performed at (TD-)DFT level to clarify the relationship existing between the (distorted) octahedral coordination of Mn(II) and the UV–Vis absorption bands. Compound **1** is proposed as a promising platform for the development of paramagnetic biomarkers.

## Figures and Tables

**Figure 1 ijms-24-07308-f001:**
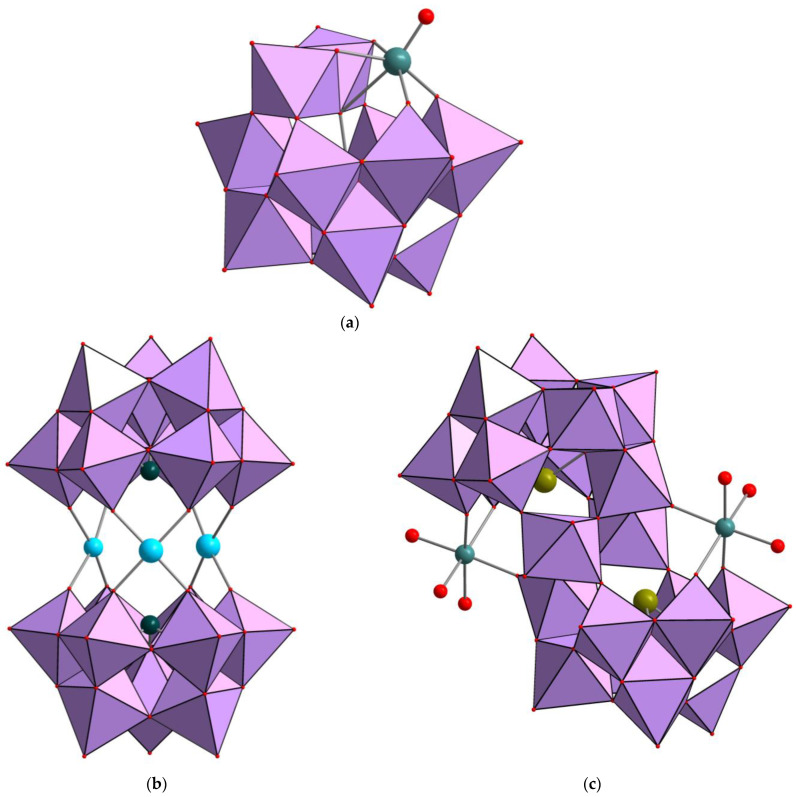
Idealized structure of [Mn(H_2_O)PW_11_O_39_]^5−^ (**a**); the structure of [(Mn^II^(H_2_O))_2.75_(WO(H_2_O))_0.25_(SbW_9_O_33_)_2_]^11.5−^, mixed Mn/W positions are shown in turquoise (**b**); the structure of [(Mn^II^(H_2_O)_3_)_2_(WO_2_)_2_(B-β-TeW_9_O_33_)_2_]^10−^. Polyhedral models based on {WO_6_} octahedral units (pink). Additional color codes: Mn—sea green; Sb—teal (**b**); Te—dark yellow (**c**); O—red.

**Figure 2 ijms-24-07308-f002:**
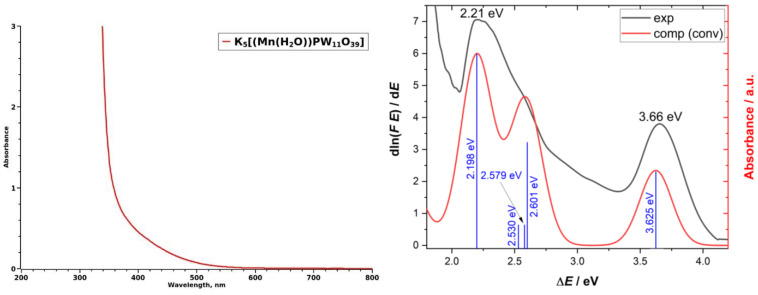
UV–Vis spectrum of **1** in water at natural pH (**left**); Experimental (black curve) and calculated (red curve) DR spectra of **1** (**right**).

**Figure 3 ijms-24-07308-f003:**
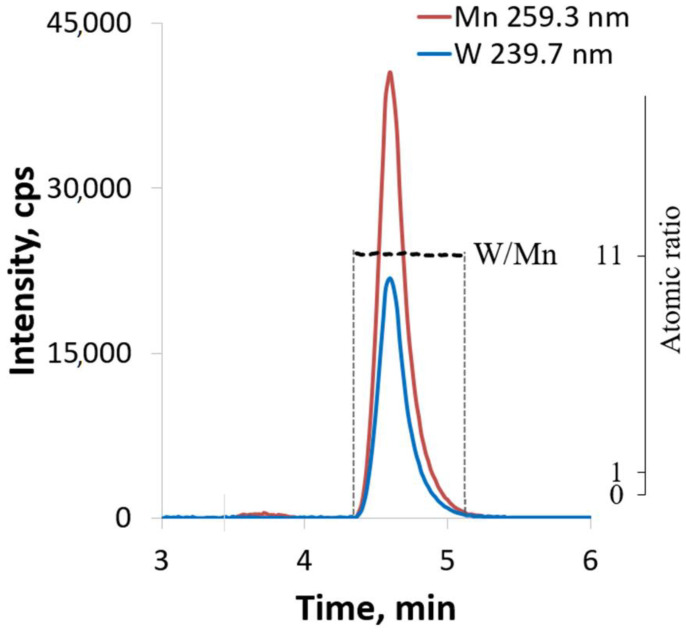
HPLC-ICP-AES chromatogram of K_5_[(Mn(H_2_O))PW_11_O_39_]·7H_2_O in water at natural pH. The lines indicate the atomic ratios of the elements (right axis) at each point of HPLC-ICP-AES chromatogram. Here and below: “Mn 259.3 nm” and “W 239.7 nm” are spectral lines of the corresponding elements used to estimate the W/Mn ratios.

**Figure 4 ijms-24-07308-f004:**
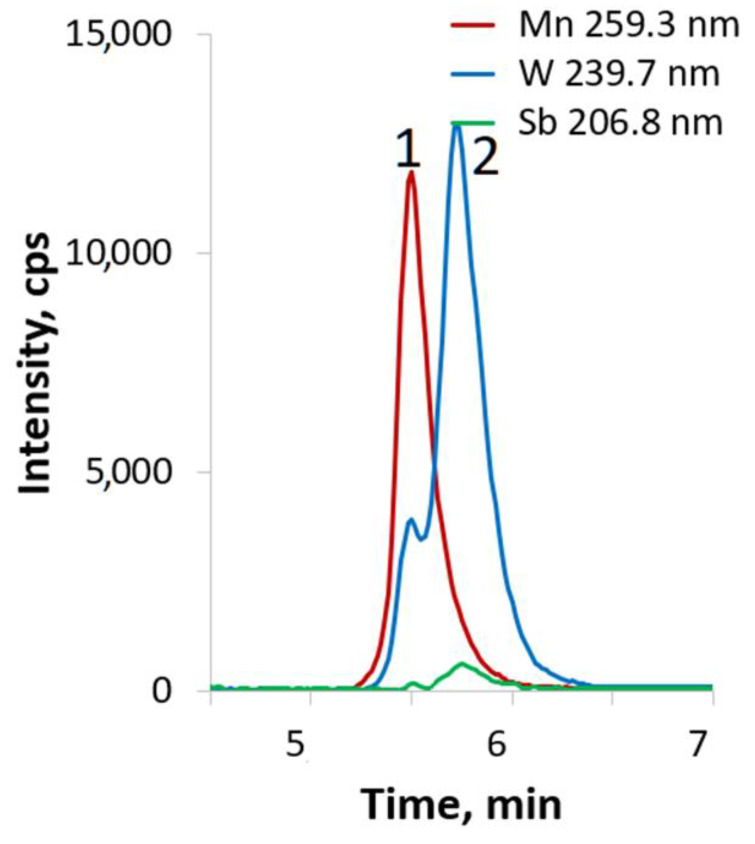
HPLC-ICP-AES chromatogram of **2** in water at natural pH.

**Figure 5 ijms-24-07308-f005:**
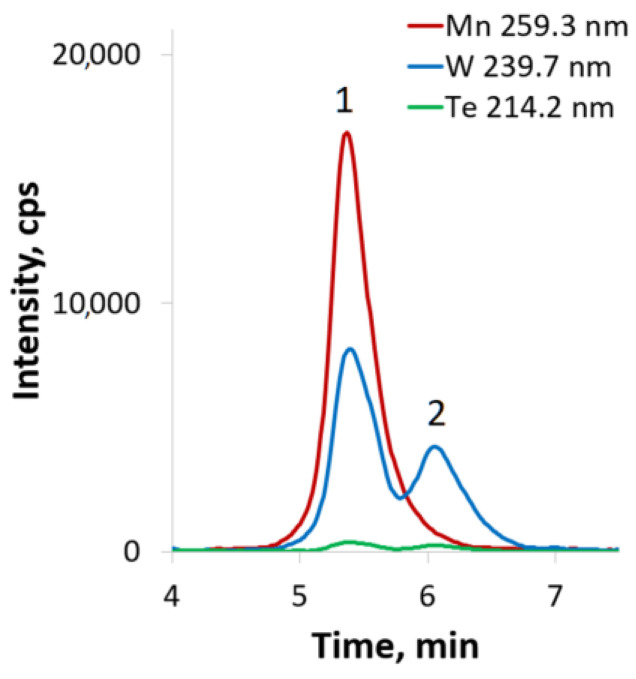
HPLC-ICP-AES chromatogram of Na_4.6_H_3.4_[(Mn^II^(H_2_O)_3_)_2_(WO_2_)_2_(β-B-TeW_9_O_33_)_2_]·19H_2_O in water at natural pH.

**Figure 6 ijms-24-07308-f006:**
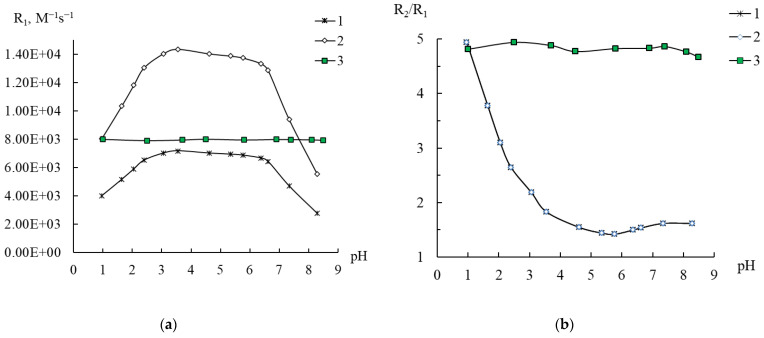
Dependence of the spin–lattice relaxivity R1 (**a**) and the ratio R_2_/R_1_ (**b**) on pH in a 0.1 M solution of **3** in water. Calculations of relaxivity were carried out taking into account manganese(II) concentrations of 0.4 mM (1) and 0.2 mM (2). For comparison, data are presented for a MnCl_2_ solution with a manganese(II) aqua ion content of 0.4 mM (3).

**Figure 7 ijms-24-07308-f007:**
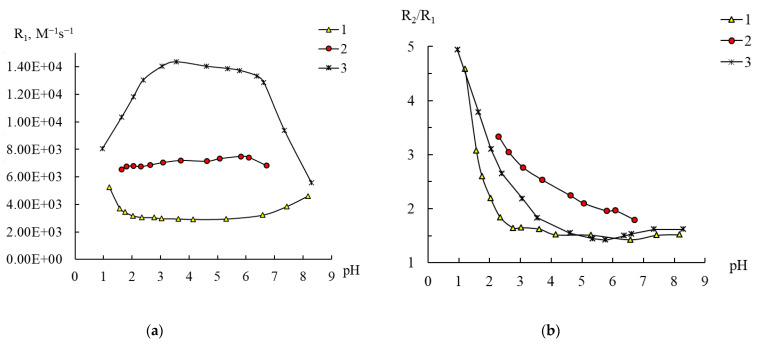
Changes in the relaxation efficiency R_1_ (**a**) and the ratio R_2_/R_1_ (**b**) on the pH of solutions of compounds **1** (0.4 mM) (1), **2** (0.133 mM) (2), and **3** (0.1 mM) (3). C_Mn(II)_ 0.2 (3), 0.4 mM (1, 2).

**Figure 8 ijms-24-07308-f008:**
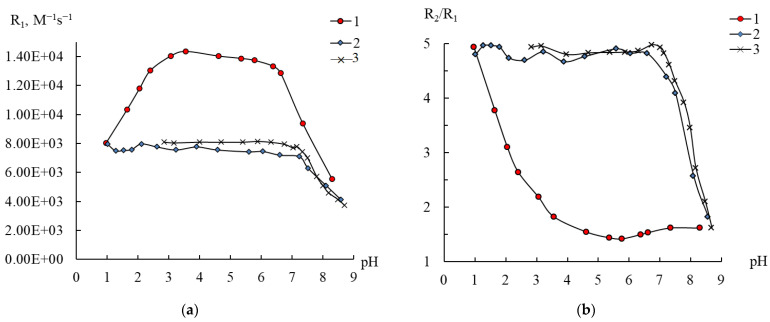
Changes in the relaxation efficiency R1 (**a**) and the R_2_/R_1_ ratio (**b**) on pH 0.1 mM solutions of compound **3** in water (1) and in PEI solution (2), and Mn(II) ions in PEI solution (3). C_Mn(II)_ 0.2 (1, 2), 0.4 mM (3), SPEI 10 mM (2, 3).

**Figure 9 ijms-24-07308-f009:**
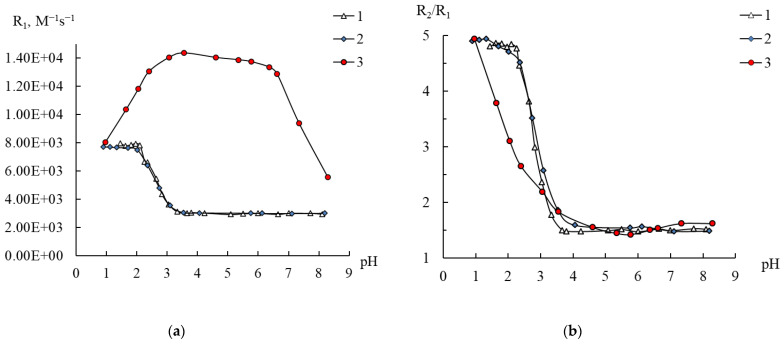
Changes in the relaxation efficiency R_1_ (**a**) and the R_2_/R_1_ ratio (**b**) as a function of the pH of Mn(II)-EDTA systems (1), compound 3—EDTA (2) aqueous solution 3 (3). C_Mn(II)_ 0.2 (2), 0.4 mM (1, 3), EDTA 1 mM (1, 2).

**Figure 10 ijms-24-07308-f010:**
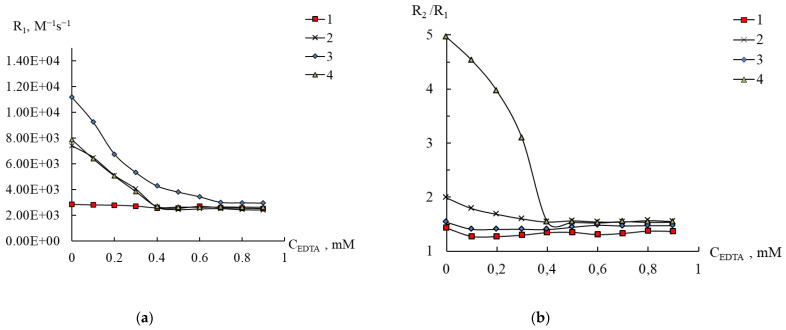
Changes in the relaxation efficiency R_1_ (**a**) and the ratio R_2_/R_1_ (**b**) on the content of EDTA in solutions of compounds **1**–**3** (1–3) and in an aqueous solution of aqua ions Mn(II) (4), C_Mn(II)_ 0.2 (3), 0.4 mM (1, 2, 4), pH 5.5.

**Figure 11 ijms-24-07308-f011:**
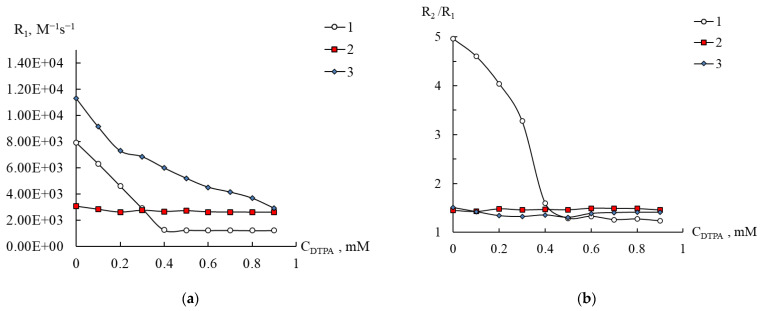
Dependence of the relaxation efficiency R1 (**a**) and the ratio R_2_/R_1_ (**b**) on the concentration of DTPA in aqueous solutions of Mn(II) (1) and compounds 1 (2) and 3 (3). C_Mn(II)_ 0.2 (3), 0.4 mM (1, 2), pH 5.5.

## Data Availability

The data presented in this study are available in Appendix A.

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
