# Peer review of "NMR-Relaxometric Investigation of Mn(II)-Doped Polyoxometalates in Aqueous Solutions"

_ijms, 2023, doi:10.3390/ijms24087308_

Round 1

Reviewer 1 Report

In the present manuscript, authors have reported the solution behavior of Mn (II)-doped polyoxometalates in aqueous solutions. State of the art NMR and HPLC-ICP-AES techniques have been used. The subject is interesting and results are discussed scientifically. The manuscript may be accepted after few minor changes.

 1) Authors have used some abbreviations which are either defined in the later part of manuscript or not defined al all e.g NMR and HPLC-ICP-AES in the abstract; MRI in line 38, DR in Line 174, NTO in line182 (defined in line 560) etc. Authors must define an abbreviation where it is used for the first time and then strictly adhere to that for clarity in text. Although few abbreviation are common e.g. NMR but even then it is better to define them in the manuscript for convenience of readers.

2) In abstract, Line 19: Sentence “According to data….” Should be revised as: According to data, the…..

3) Heading of subsection 2.1needs revision

4) It’s better to split subsection 2.2 in two different headings i.e  “Electronic absorption spectra” and “Quantum-chemical calculations of Mn2+ electronic states” for clarity

5) Correct the typo in line 205: “Therefore”

6) Why effect of polyelectrolyte is investigated only for compound 3? What was the criteria for selection of compound 3 for this specific investigation?

7) % purities of the compounds used should be given in subsection 3.1.

8) In conclusions section line 574: Sentence “it was used an approach based on the analysis of the ratio of relaxivities R2/R1” needs revision

Author Response

In the present manuscript, authors have reported the solution behavior of Mn (II)-doped polyoxometalates in aqueous solutions. State of the art NMR and HPLC-ICP-AES techniques have been used. The subject is interesting and results are discussed scientifically. The manuscript may be accepted after few minor changes.

 1) Authors have used some abbreviations which are either defined in the later part of manuscript or not defined al all e.g NMR and HPLC-ICP-AES in the abstract; MRI in line 38, DR in Line 174, NTO in line182 (defined in line 560) etc. Authors must define an abbreviation where it is used for the first time and then strictly adhere to that for clarity in text. Although few abbreviation are common e.g. NMR but even then it is better to define them in the manuscript for convenience of readers.

Thank you for the comment! We tried to define all non-standard abbreviations in the text.

2) In abstract, Line 19: Sentence “According to data….” Should be revised as: According to data, the…..

Thank you very much! This was corrected!

3) Heading of subsection 2.1needs revision

Thank you very much! This was corrected!

4) It’s better to split subsection 2.2 in two different headings i.e  “Electronic absorption spectra” and “Quantum-chemical calculations of Mn2+ electronic states” for clarity

Thank you very much! This was corrected!

5) Correct the typo in line 205: “Therefore”

Thank you very much! This was corrected!

6) Why effect of polyelectrolyte is investigated only for compound 3? What was the criteria for selection of compound 3 for this specific investigation?

Compound 3 has the highest relaxivities in aqueous solution compared to the other two polyoxometalates. It was previously found that the binding of manganese complexes MnL (L = EDTA, DTPA) with PEI leads to an increase in relaxivity due to the slowdown in the rotation of such Mn-L-PEI associates (see discussion in section 2.4, lines 275-287, the reasons for the increased relaxivity values of the compound 3 compared to the others, since for manganese the relaxation times are controlled by the correlation rotation time τR). In this connection, it was of interest whether it is possible to achieve an additional increase in the relaxivity of compound 3 upon its interaction with a cationic polymer. However, it turned out that polyoxometalate is completely destroyed with the release of manganese ions.

7) % purities of the compounds used should be given in subsection 3.1.

The corresponding information has been added to the subsection 3.1.

8) In conclusions section line 574: Sentence “it was used an approach based on the analysis of the ratio of relaxivities R2/R1” needs revision

Thank you very much! This was corrected!

Reviewer 2 Report

The authors reported on the Mn(II)-doped polyoxometalates among three polyoxometalate complexes that exhibit high relaxivity values using NMR relaxometry. The study showed that the relaxivity values of these materials can be modified by changing their composition and concentration in solution. The authors also found that the presence of additives in solution can affect the magnetic properties of these materials, which could have implications for their use in biomedical applications.

Polyoxometalates, emerging inorganic metal-oxide clusters, are promising candidates for the development of nanoprobes due to their biological activity in theranostics and MRI. The article are well-written and organised. I, therefore, recommend the publication on International Journal of Molecular Sciences upon the following conditions are well addressed.

[1] In Figure 1, please indicate clearly that for which purple tetrahedra, differently coloured spheres represent for.

[2] There are some abbreviations not mentioned in the manuscript or some terminologies not defined clearly such as “DR spectrum”, “Keggin anion” and “NTO analysis”.

[3] Please show the details of how the DFT performed, especially for the NTO analysis.

[4] What is the purpose for the DR spectrum?

[5] In Figure 3, it is not clear that where are the obtained values from for the peak positions of Mn and W. “Mn 259.3 nm” and “W 239.7 nm”

[6] In the manuscript, the authors did not clearly describe how the presence of additives in solution can affect relaxivity values of the Mn(II)-doped polyoxometalates.

[7] In Figure S1, what is the method to calculate the X-ray powder diffraction pattern of the compound 2?

[8] In Figure S2, please indicate clearly that for which the differently coloured spheres represent for.

Author Response

The authors reported on the Mn(II)-doped polyoxometalates among three polyoxometalate complexes that exhibit high relaxivity values using NMR relaxometry. The study showed that the relaxivity values of these materials can be modified by changing their composition and concentration in solution. The authors also found that the presence of additives in solution can affect the magnetic properties of these materials, which could have implications for their use in biomedical applications.

Polyoxometalates, emerging inorganic metal-oxide clusters, are promising candidates for the development of nanoprobes due to their biological activity in theranostics and MRI. The article are well-written and organised. I, therefore, recommend the publication on International Journal of Molecular Sciences upon the following conditions are well addressed.

[1] In Figure 1, please indicate clearly that for which purple tetrahedra, differently coloured spheres represent for.

Thank you for the comment. The corresponding color codes have been added to the revised version of MS.

[2] There are some abbreviations not mentioned in the manuscript or some terminologies not defined clearly such as “DR spectrum”, “Keggin anion” and “NTO analysis”.

Thank you for the comment! We tried to define all non-standard abbreviations in the text.

[3] Please show the details of how the DFT performed, especially for the NTO analysis.

Upon geometry optimization, carried out at DFT B3LYP[GD3BJ] / 6-311++G** (for H, P, O, and Mn) & LANL2TZ(f) +ECP (for W) level (see Computational Details), one of the two H atoms of the water molecule interacts with an O atom of the cluster (H-Oclus = 2.268 Å), whereas the coordination of the Mn2+ was found to be distorted prolate octahedron.

Time-dependent density-functional theory (TD-DFT) calculations were performed using ω-B97X[D] functional and the same basis sets used for the geometry optimisation to study the nature of the yellow coloration associated with Mn2+ into the cavity of [PW11O39]7–.

All details are shown in the 3.8. section

[4] What is the purpose for the DR spectrum?

The purpose was to extract the electronic transition more carefully. The regular UV-VIS data are not so informative.

[5] In Figure 3, it is not clear that where are the obtained values from for the peak positions of Mn and W. “Mn 259.3 nm” and “W 239.7 nm”

Mn 259.3 nm and W 239.7 nm are spectral lines used to estimate the W/Mn ratios. This was added to the figure capture.

[6] In the manuscript, the authors did not clearly describe how the presence of additives in solution can affect relaxivity values of the Mn(II)-doped polyoxometalates.

It was previously found that the binding of manganese complexes MnL (L = EDTA, DTPA) with PEI leads to an increase in relaxivity due to the slowdown in the rotation of such Mn-L-PEI associates (see discussion in section 2.4, lines 271-277, the reasons for the increased relaxivity values of the compound 3 compared to the others, since for manganese the relaxation times are controlled by the correlation rotation time τR). In this connection, it was of interest whether it is possible to achieve an additional increase in the relaxivity of compound 3 upon its interaction with a cationic polymer. However, it turned out that polyoxometalate is completely destroyed with the release of manganese ions.

Studies of the effect of chelating agents should have revealed the stability of polyoxometalates in the presence of strong complexing agents. This study was carried out based on the difference in the relaxivities of solutions of compounds 1-3 in water and in solutions of EDTA, DTPA. Unfortunately, at this stage, it was not possible to explain why compound 2 requires an excess of complexone for complete extraction of manganese from compound 2.

[7] In Figure S1, what is the method to calculate the X-ray powder diffraction pattern of the compound 2?

We used the structural model refined from the single crystal X-ray diffraction data to calculate the corresponding powder pattern using Mercury program.

[8] In Figure S2, please indicate clearly that for which the differently coloured spheres represent for.

Thank you for the comment. The corresponding color codes have been added to the revised SI file.
